# The Relationship Between Climate Change and Breast Cancer and Its Management and Preventative Implications in South Africa

**DOI:** 10.3390/ijerph22101486

**Published:** 2025-09-25

**Authors:** Pululu Sexton Mahasa, Muambangu Jean Paul Milambo, Sibusiso Frank Nkosi, Geofrey Mukwada, Martin Munene Nyaga, Solomon Gebremariam Tesfamichael

**Affiliations:** 1Division of Public Health, Faculty of Health Sciences, University of the Free State, Bloemfontein 9300, South Africa; jeanpaulmilambo2@gmail.com (M.J.P.M.); mpumalangaepidemiologyresearch@gmail.com (S.F.N.); 2Department of Geography, University of the Free State, Qwaqwa Campus, Phuthaditjhaba 9866, South Africa; gmukwada@gmail.com; 3Department of Gynaecology and Obstetrics, Faculty of Medicine and Health Sciences, Walter Sisulu University, Mthatha 5117, South Africa; 4Mpumalanga Department of Health, Carolina 1185, South Africa; 5Next Generation Sequencing Unit, Division of Virology, School of Biomedical Sciences, Faculty of Health Sciences, University of the Free State, Bloemfontein 9300, South Africa; nyagamm@ufs.ac.za; 6Department of Geography, Environmental Management and Energy Studies, University of Johannesburg, Johannesburg 2006, South Africa; sgtesfamichael@uj.ac.za

**Keywords:** breast cancer, climate change, UV radiation, rising temperature, air pollution, endocrine-disrupting chemicals (EDCs), breast cancer, pesticides, phthalates, p,p’-dichlorodiphenyltrichloroethane (DDT), polychlorinated biphenyls (PCBs), p,p’dichlorodiphenyldichloroethylene (DDE), 2,4-dichlorophenoxyacetic acid, 2,3,7,8-tetrachloridibenzo-p-dioxin (TCDD or Dioxin), polybrominated diphenyl ethers (PBDEs), oral contraceptive pills, resveratrol (RES)

## Abstract

This review aims to explore the implications of climate change for breast cancer management and prevention, with a focus on global strategies and interventions that can be applied in various contexts, including South Africa. Climate change has emerged as a significant global health concern, with far-reaching implications for various diseases, including cancer. This systematic review aims to synthesise epidemiological research examining the relationship between climate change and the incidence of breast cancer. We conducted a comprehensive literature search using main search terms, including “breast cancer,” “climate change,” “air pollution,” “water pollution,” “global warming,” and “greenhouse effect,” supplemented by the general term “breast” cancer across multiple databases. Our analysis identified studies that link environmental changes—such as rising temperatures, altered precipitation patterns, and increased exposure to pollutants—with breast cancer risk. Our findings highlight a potential association between climate-related factors, including heat stress, air and water pollution, endocrine-disrupting chemicals, and lifestyle changes influenced by environmental shifts, and the epidemiology of breast cancer. This review underscores the need for an integrated approach that incorporates climate science into public health strategies to mitigate breast cancer risk. By elucidating these connections, we aim to inform policymakers and healthcare professionals about the importance of addressing climate change not just as an environmental issue, but as a pressing determinant of health that may exacerbate cancer incidence, particularly in vulnerable populations. Further research is warranted to elucidate the underlying biological mechanisms and to develop targeted interventions that can address both climate change and its potential health impacts.

## 1. Introduction

This review aims to explore the implications of climate change for breast cancer management and prevention, with a focus on global strategies and interventions that can be applied in various contexts, including South Africa. Climate change is an ever-changing phenomenon that may negatively impact human health, possibly leading to breast cancer [1,2,3]. Internationally, breast cancer is one of the most diagnosed cancers worldwide and ranks as the most prevalent cancer among women [4,5]. In terms of global cancer incidence, breast cancer typically ranks first or second, depending on factors such as geographical location, age distribution, and access to healthcare [4,6,7,8]. It is also important to note that breast cancer incidence rates may vary among different populations and regions [9,10,11]. Breast cancer incidence varies over the world, with Western countries having a greater incidence than Eastern ones [4,5,6,8,9]. But in low- and middle-income nations, the incidence of breast cancer is rapidly increasing because of dietary changes, lifestyle modifications, and increased longevity [12].

Numerous studies have demonstrated the correlation between several factors and an increased risk of breast cancer [13,14,15,16,17,18,19,20,21,22,23,24,25,26,27,28,29,30,31,32,33,34,35,36,37,38,39,40,41,42,43,44,45,46,47,48,49,50,51,52,53,54,55,56,57,58,59,60,61,62,63,64,65,66,67,68,69,70,71,72,73,74,75,76,77,78,79,80,81,82,83,84,85,86,87,88,89,90,91,92,93,94,95,96,97,98,99,100,101,102,103,104,105,106,107,108,109,110,111]. The factors influencing breast cancer risk encompass a variety of elements, including age, race, and socio-economic status, along with genetic factors like BRCA mutations [13,14,15,16,17,18,19,20,21,22,23,24,25,26,27,28,29,30,31,32,33,34,35,36,37,38,39,40,41,42,43,44,45,46,47,48,49,50,51,52,53,54,55,56,57,58,59,60,61,62,63,64,65,66,67,68,69,70,71,72,73,74,75,76,77,78,79,80,81,82,83,84,85,86,87,88,89,90,91,92,93,94,95,96,97,98,99,100,101,102,103,104,105,106,107,108,109,110,111]. Hormonal factors such as parity, age at menarche, and age at first full-term pregnancy, as well as nursing, also play a role [17,21,23,28]. Additionally, lifestyle factors, including physical activity, diet, alcohol consumption, and tobacco use, are significant contributors to breast cancer risk [36,100]. Breast cancer consistently remains a significant public health concern due to its high incidence and impact on morbidity and mortality [7,8].

Over the past few decades, there has been a notable increase in global temperatures, along with a rise in the frequency and intensity of climate change-related phenomena [3,112]. These include extreme weather events, such as prolonged heatwaves and devastating wildfires, as well as sand and dust storms (SDSs) and severe thunderstorms [113,114,115,116]. Such events not only pose immediate threats to ecosystems and human health but also serve to exacerbate global warming itself, creating a feedback loop that could lead to even more severe climate issues in the future [3,112]. Among the various fields of study emerging in the wake of these environmental challenges, the potential link between climate change and breast cancer has garnered increased attention and represents an area of research that certainly warrants further investigation [117]. Although direct causality between the two has yet to be firmly established, several indirect pathways have been proposed that suggest potential associations between climate change and an elevated risk of breast cancer [118,119,120]. These pathways include various environmental exposures to harmful substances, significant changes in lifestyle behaviours as a result of climate-related factors, and the impact of climate change on food supply and nutritional availability, as well as increased psychosocial stress levels induced by the uncertainties and challenges posed by climate change [119,120,121]. Such complex interactions underscore the importance of addressing climate change not only as an environmental issue but also as a potential public health concern, necessitating a multidimensional research approach to comprehensively understand the implications for breast cancer risk and overall health outcomes [3,122].

Climate change can result in significant shifts in environmental conditions, leading to adverse impacts such as increased air pollution, water contamination, and heightened exposure to endocrine-disrupting chemicals [122,123,124,125,126,127]. Prolonged exposure to these environmental pollutants has been linked to an elevated risk of developing breast cancer, as these substances have the potential to interfere with normal hormonal functions and contribute to carcinogenic processes in the body [128]. Moreover, climate change-related events, including extreme weather incidents and natural disasters, can disrupt communities at multiple levels [129,130]. These disruptions can lead to changes in lifestyle behaviours that may further influence health outcomes [131,132]. For example, communities affected by such events often experience shifts in diet due to food supply shortages, physical activity levels may decrease as safe spaces for exercise become limited, and access to healthcare services can be compromised in the aftermath of disasters. These lifestyle changes can significantly influence breast cancer risk factors, such as obesity, which is known to be associated with increased breast cancer incidence [133,134]. Additionally, hormonal balance may be disrupted due to variations in diet and stress levels resulting from the unstable environmental conditions brought on by climate change [135,136,137]. The interplay of these various factors highlights the complex relationship between environmental conditions, lifestyle adjustments, and health outcomes, emphasising the need for a proactive approach to mitigate the impacts of climate change on public health, particularly regarding cancer risk [138,139].

Climate change has the potential to significantly disrupt agricultural systems, which can lead to profound changes in the availability, quality, and nutritional content of the food supply [140]. As environmental conditions shift, we may see variations in crop yields and food production practices, resulting in shifts in dietary patterns that may be less healthful [139,140]. The increased consumption of processed or contaminated foods, which may occur in response to these disruptions, can have detrimental effects on health and may particularly impact breast cancer risk [141,142]. In addition to the direct effects on food systems, climate change-related events—including severe heatwaves, devastating wildfires, and the displacement of communities due to rising sea levels—can cause substantial psychosocial stress and lead to various breast cancer and mental health issues [3,10,99]. The stress of coping with such challenges can extend beyond the immediate impact, influencing individuals’ long-term emotional and psychological well-being [129,130].

Chronic stress is known to have significant breast cancer physiological consequences [143,144]. It has been linked to immune dysregulation, increased inflammation, and altered hormone levels, all of which may influence breast cancer progression and treatment response [145,146]. For instance, sustained high stress levels can negatively affect the body’s immune response, making it less effective in detecting and combating breast cancerous cells [147]. As a result, the interplay between environmental factors and mental health is crucial for understanding how climate change may indirectly contribute to breast cancer risk [147,148]. This highlights the urgent need for integrated approaches that consider not just physical health, but also mental well-being in the context of climate-related changes, as both are critical in determining health outcomes in affected populations [149,150,151].

While these pathways suggest potential links between climate change and breast cancer, further research is needed to elucidate the precise mechanisms and establish causality [122,152,153]. Interdisciplinary studies integrating environmental science, epidemiology, oncology, and public health are essential for understanding the complex relationship between climate change and breast cancer risk and informing evidence-based interventions for prevention and mitigation [3,131].

The purpose of this review is to examine the literature on the connection between breast cancer and climate change, as well as any implications for treatment and prevention.

## 2. Methods

### 2.1. Literature Search Strategy

This scoping review adhered to the Joanna Briggs Institute (JBI) guidelines, which include creating review questions, creating a protocol, creating eligibility standards, search tactics, creating exclusion and inclusion criteria, choosing studies, extracting data, and analysing data. Additionally, the review followed the reporting guidelines outlined in the Meta-Analyses Extension for Scoping Reviews and the Preferred Reporting Items for Systematic Reviews (PRISMA).

Following an initial search on EBSCOhost, adjustments were made to extend the search to additional databases, including Web of Science, Cochrane, and Scopus. The EBSCOhost databases to be included are Africa-Wide Information, Academic Search Ultimate, APA E-Journals, GreenFILE, CAB Abstracts with Full Text, CINAHL with Full Text, PsycINFO, Applied Science & Technology Source Ultimate, Health Source: SPORTDiscus with Full Text, MEDLINE, and Nursing/Academic Edition. Only peer-reviewed English-language publications from each database’s founding until 21 February 2025 were taken into account. The primary search terms included “greenhouse effect,” “global warming,” “air pollution,” “water pollution,” “breast cancer,” and “climate change,” with the broad term “breast” cancer incorporated.

All documents retrieved during the search were collected and organised in the Mendeley Reference Manager (Version v2.137.0) for further evaluation. Duplicate entries were removed using the software’s built-in functionality. Titles, abstracts, and full texts were screened using a double-blind process based on established inclusion criteria, and the reasons for excluding certain papers were documented.

The search strategy employed the following keywords:

(“Environmental Exposure” OR “Climat* change*” OR “Air pollut*” OR “Water pollut*” OR “global warming” OR “Greenhouse effect*” OR “Greenhouse gas*”) AND ti (“breast neoplasm*” OR “Breast cancer*” OR “breast tumour*” OR “breast tumor*”).

Only peer-reviewed articles written in English that were published between the database’s launch and 21 February 2025 were included. Breast cancer, climate change, air pollution, water pollution, global warming, and greenhouse effect were the main search terms used. The general word “breast” cancer was used in the search terms.

The PROSPERO (International Prospective Register of Systematic Reviews) study record is CRD42025625376.

### 2.2. Study Selection

Every document found during the search was compiled and added to the Mendeley Reference Manager for review. Using a feature integrated into the software, duplicates were eliminated. A double-blind approach was used to screen titles, abstracts, and complete texts according to predetermined inclusion criteria (Table 1). There was documentation of the papers’ exclusion reasons.

### 2.3. Flowchart and Scope of Review

A flowchart was created to classify the effects of climate change on breast cancer (Figure 1). This was developed over the course of the review using an iterative approach that began with reading the literature. The graphic served as both an organising structure for presenting the findings and a means of outlining the review’s scope.

### 2.4. Data Extraction

Information regarding the participants, study topics, context, techniques, publication journal, subject area, ranking, and noteworthy discoveries were among the data that were extracted. To ensure accuracy, the data extraction method was developed and tested on two publications. The extractor initially reviewed the study objectives to identify information relevant to the elements of the causal pathway. In cases where a study could not conclusively identify a focus related to the causal pathway, relevance was determined by analysing the study procedures and outcomes. The publication type of each paper was classified according to the journal’s system, with all non-primary research papers—such as reviews, opinions, editorials, perspectives, and essays—termed “secondary research papers.”

### 2.5. Data Analysis and Synthesis

The iteratively created causal pathway was used to assess the data deductively (Figure 1). Every reviewer talked about the data that were extracted, how they related to the causal route, and which areas needed more research. Three forms were used to present the data: tabular, narrative, and diagrammatic, to demonstrate how the studies that were included discussed how climate change affects breast cancer.

## 3. Results

The search resulted in 1121 publications. Following the removal of duplicates, screening of titles and abstracts, and a full-text review, 232 publications were included in this review (Figure 1). Figure 1 is a flowchart of the literature search and selection process based on the PRISMA Extension for Scoping Reviews adapted from Watson et al., 2024 [131].

### 3.1. Publication Characteristics

A total of 1121 publications were identified through a database search, which included various types of studies, such as 158 case–control studies, 65 primary studies, 6 case studies, and 3 cross-sectional studies. After eliminating 89 duplicate entries, 218 publications were not reviewed due to the unavailability of full texts or because the full texts were not published in English. This initial screening narrowed down the pool of relevant documents significantly.

Additionally, 582 publications were excluded from consideration for various reasons. Exclusions included research protocols, theses, conference abstracts, books, and articles that did not provide substantial content, such as interviews or letters to the editor. Moreover, some publications were excluded for discussing issues related to ozone depletion rather than climate change, and others lacked references to primary causative factors, which were essential for the intended analysis. This rigorous exclusion process aimed to refine the dataset to include only those studies pertinent to the objectives of the research.

The review of the literature related to the specified topic highlights a diverse array of primary research originating from multiple countries across various continents. This global representation, including nations such as Australia, Belgium, and the Czech Republic, underscores the widespread interest and investment in the research area, reflecting unique geographical, cultural, and socio-political contexts that may influence the findings. For instance, studies from India and China may provide insights into climate challenges in densely populated regions, while research from the United States or the United Kingdom might focus on advanced technological responses to these challenges.

The inclusion of papers from countries like Brazil, Mexico, and South Africa also emphasises the importance of understanding the interplay between climate change and socio-economic factors in developing nations. Research from South Korea and Europe demonstrates an increasing commitment to sustainability and advancement in climate policies, further enriching the data through varied methodologies and local experiences. This international pool of data not only enhances the comprehensiveness of the review but also allows for cross-comparison of results, contributing to a more nuanced understanding of global trends and local responses to climate-related issues. By drawing on insights from such a wide array of countries, the review can better elucidate the complex ways in which different regions confront climate change and associated health impacts, ultimately informing more effective global strategies for adaptation and mitigation.

Of the 158 case–control research articles, 152 were published in health-related Q1-ranked journals (by SCImago definition being in the top 25%), 65 were primary studies, 6 were case studies, 3 were cross-sectional studies, and only 9 reviews were published in Q1 journals. Of the secondary research papers, none followed established methodological guidelines for conducting a review, although ten provided limited descriptions of their review methods or search strategies. Thirteen of the primary research articles focused on quantitative analyses of atmospheric variables and health outcomes, one combined qualitative and quantitative analyses, and two were based on a cross-sectional survey.

### 3.2. Conceptual Framework

Figure 2, an illustration of the conceptual framework, visually depicts the ongoing and intensifying relationship between climate change and breast cancer incidence, providing a simplified representation of the complex interactions identified in the literature review. This diagram aims to facilitate a clearer comprehension of the scope and complexity of the topic, but it is essential to acknowledge that it only highlights key connections and does not purport to be an exhaustive illustration of the intricate relationships between breast cancer risk factors and climate change.

One significant factor highlighted is the influence of rising ambient temperatures on behaviour, particularly regarding exposure to ultraviolet (UV) radiation. As temperatures increase, behaviours associated with outdoor work and recreational activities are likely to contribute to elevated levels of UV exposure among individuals [97,98,99]. Research has established a complex inverse U-shaped relationship between changing behaviours and ambient temperatures; specifically, adults exhibit a two- to three-fold increase in the likelihood of experiencing sunburn when temperatures exceed 22 °C. However, it is also observed that in extreme heat conditions, discomfort may prompt people to spend more time indoors, potentially reducing their UV exposure.

Furthermore, individuals working in rural settings, where outdoor occupations like farming and forestry are prevalent, may face even higher exposure to UV radiation due to prolonged time spent outdoors. This increased risk is particularly concerning when considering the rising incidence of breast cancer documented in both human studies and animal models. Research has indicated that heightened ambient temperatures—independent of UV exposure—are being investigated as a contributing factor to the increasing cases of breast cancer.

Additionally, the potential role of air pollution in the escalation of breast cancer incidence has been a focus of recent studies. As climate change exacerbates air quality issues, understanding how air pollutants interact with biological systems and potentially increase cancer risk becomes critical. This multifaceted relationship underscores the importance of further investigation into how environmental factors related to climate change can influence breast cancer risk and necessitates a comprehensive approach to public health strategies aimed at mitigating these risks.

### 3.3. Summary of Findings

Of the 232 included papers in Appendix A, 68 examined UV radiation as a cause of breast cancer; 43 examined shifts in behaviour related to climate change as a cause of breast cancer; 21 examined rising temperatures as a risk factor for breast cancer on its own; 11 examined occupational, rural, or other contextual factors as a risk factor for breast cancer; 7 examined the impact of climate change on prevention and access to care for breast cancer; and 5 examined air pollution and its possible association with breast cancer. Six of the secondary research papers particularly addressed breast cancer and climate change. The remaining ones covered topics such as dermatology, cancer, and general health with a broader focus.

#### 3.3.1. Pesticides and Endocrine-Disrupting Chemicals (ECDs) as a Cause of Breast Cancer

A total of 122 studies have investigated whether breast cancer patients exhibit higher concentrations of specific pesticides in their blood compared to healthy women without serious illnesses. This inquiry is particularly significant, as pesticides such as DDT and its metabolites (DDD and DDE), dieldrin, heptachlor, and hexachlorocyclohexane (HCH) and its isomers (alpha, beta, and gamma) are known to be lipid-soluble and non-biodegradable and to act as endocrine disruptors [154,155]. These pesticide characteristics raise concerns regarding their potential to contaminate both living organisms and environmental components [156]. Pesticides that mimic oestrogen or are linked to breast cancer development in animal studies are of particular interest when exploring breast cancer aetiology [157,158,159,160]. Despite these investigations, prior research has faced limitations, including the assessment of pesticide residues in tissues at a single point in time and the focus on a limited number of pesticides that are no longer in widespread use globally, particularly in agriculturally intensive countries [161,162].

Compounds exhibiting oestrogenic activity, often classified as xenoestrogens, encompass a broad range of substances capable of mimicking oestrogen in the body. Agricultural pesticides such as p,p’-Dichlorodiphenyltrichloroethane (DDT), p,p’-Dichlorodiphenyldichloroethylene (DDE), atrazine, and 2,3,7,8-Tetrachlorodibenzo-p-dioxin (commonly referred to as TCDD or dioxin) are notable examples [161,162]. Additionally, synthetic chemicals play a critical role among xenoestrogens, including BPA, phthalates, per- and polyfluoroalkyl substances (PFAS), parabens, polychlorinated biphenyls (PCBs), and polybrominated diphenyl ethers (PBDEs), along with hormonal contraceptives [131]. These substances are prevalent in many consumer products and industrial applications, contributing to widespread environmental exposure [15,131].

Beyond synthetic compounds, natural substances known as phytoestrogens, such as genistein and resveratrol, also mimic oestrogenic effects [99,154]. Additionally, certain mycotoxins, particularly zearalenone, are significant xenoestrogens that pose risks to human health and animal reproduction [32,102]. The coexistence of both synthetic and natural xenoestrogens in our environment raises critical concerns regarding their potential endocrine-disrupting effects, disrupting normal hormonal functions and contributing to various health issues, including reproductive disorders and specific types of cancers [106,110,155]. Understanding the sources and impacts of these compounds is essential for informing effective regulatory measures and public health strategies to mitigate their potential health impacts [109,154].

#### 3.3.2. Workplace, Rural Settings, and Other Elements Influencing Breast Cancer Incidence

A total of 31 studies have investigated environmental, occupational, or rural characteristics that may contribute to an elevated risk of breast cancer [163]. One notable feature was that eleven studies clearly alighted on work settings that could have given rise to breast cancer incidences, whereas twenty-three pointed to rurality alone. The findings from Neale et al. (2023) are particularly noteworthy, as they concluded that outdoor workers face a higher risk of developing breast cancer based on their comprehensive epidemiological analysis [163,164]. This connection underscores the importance of considering occupational exposures in breast cancer risk assessments, as the work environment can play a crucial role in overall health outcomes [164].

Furthermore, Silva and Rosenbach (2021) indicated that labourers with lower incomes are more likely to be employed in outdoor settings, suggesting a correlation between socio-economic status and exposure risk [164]. This relationship points to broader systemic issues, where marginalised groups may face increased breast cancer risks due to their occupational circumstances [165,166,167,168]. Compounding this situation of marginalised population groups being exposed to increased breast cancer risk, research has shown that populations with lower socio-economic status often present with aggressive breast cancer as assessed through a combination of histopathological characteristics, molecular markers, stage, and the tumour’s behaviour in response to treatment, which is indicative of later-stage diagnosis and potentially poorer health outcomes [169,170,171,172]. Cassidy et al. 2025 [169] and Cupertino et al. 2025 [173] strongly highlight the critical need for targeted public health interventions that address the disparities in breast cancer risk and promote early detection strategies, particularly among vulnerable populations who may be disproportionately affected by environmental and occupational hazards.

#### 3.3.3. Rising Temperature in Isolation as a Cause of Breast Cancer

A total of 21 studies have investigated rising temperature in isolation as a cause of breast cancer. The exploration of ambient temperature as a possible factor affecting breast cancer risk has been investigated through both original research and various secondary publications [174]. Specifically, two original research studies and nineteen secondary research articles focused on examining how climate change and rising temperatures could be linked to an increase in breast cancer incidence. This growing body of literature reflects heightened scientific interest in the intersections between environmental factors and public health outcomes [175,176,177,178,179]. As climate change continues to exacerbate global temperatures, understanding its implications for cancer risk has become increasingly critical, prompting researchers to investigate the potential for environmental modifications to influence cellular behaviour, immune response, and hormonal regulation—all of which are key factors in the development of breast cancer [175,180,181].

For instance, a pivotal study by Van der Leun and de Gruijl (2002) [176] established a direct correlation between rising temperatures and increased incidence rates of keratinocyte carcinoma, a type of breast cancer primarily influenced by UV exposure. This finding positions temperature as a significant risk factor in the context of skin malignancies, raising questions about whether similar mechanisms might apply to breast cancer; meanwhile, growing evidence is emerging [175,182,183]. The underlying biological processes—such as the impact of heat on gene expression, inflammation, and metabolic shifts—could operate at a cellular level, potentially affecting breast tissue and tumour biology. These insights into the relationship between environmental stressors and cancer highlight the need for further investigation into how rising temperatures might influence the risk of breast cancer through various pathways.

In contrast, two review publications expressed uncertainty regarding the exact mechanisms through which ambient temperature affects breast cancer risk. These reviews underscore the complexity of the relationship between climate change and health outcomes, noting that while there may be correlations, causation remains difficult to establish unequivocally. Additionally, factors such as individual susceptibility, the interplay of various environmental exposures, and lifestyle choices complicate the understanding of how ambient temperature specifically contributes to breast cancer risk. The lack of consensus in the current literature highlights the necessity for more comprehensive studies that consider multifaceted variables and include diverse populations. Understanding these complexities is essential for developing effective public health strategies aimed at mitigating the risks associated with climate change and for ensuring equitable health outcomes across different demographic groups.

#### 3.3.4. Behaviours Related to Temperature and Exposure to UV Radiation and Occurrence of Breast Cancer

A total of 14 studies (Appendix A) has investigated behaviours related to temperature and exposure to UV radiation and occurrence of breast cancer. The discussion surrounding ultraviolet (UV) radiation emphasises the critical importance of public health recommendations aimed at understanding and managing exposure levels [17,184,185,186,187,188,189,190,191,192,193,194,195,196,197,198,199,200,201,202,203,204,205,206,207,208,209,210,211,212,213,214,215,216,217,218,219,220,221,222,223,224,225,226,227]. Several studies have directly examined the relationship between personal UV radiation (UVR) exposure and breast cancer incidence. However, findings have been inconsistent, supporting either a reduced risk [184,185,186,187,188,189] or no association [17,190,191,192,193]. UVR exposure has been estimated in prior studies using a variety of metrics, most commonly either ambient UVR or time outdoors. Many studies reporting an association between UVR and breast cancer have used time outdoors as the metric of UVR exposure [184,186,189,192], and some have evaluated ambient UVR exposure [17,187,189,192]. Overall, few studies investigating relationships between UVR-related factors and risk of breast cancer have been prospective [184,185,186,189,192,193], and only a few of these have included a wide range of ambient UVR exposures over the course of a lifetime [17,186,187,192,193,195]. The following table is attached as a Appendix A in this article.

In Appendix A, we present papers included in the study, and these tables are attached as appendices to this article.

Various publications have underscored the need for clear and accessible guidelines that enable individuals to assess their personal UV exposure accurately [196,197,198,199,200,201,202,203,204,205,206,207,208,209,210,211,212,213,214,215,216,217,218,219,220,221,222,223,224,225,226,227]. By empowering people with knowledge about UV radiation levels, these guidelines facilitate informed decision-making regarding sun safety measures, such as when to seek shade or apply sunscreen [198,199,211,212,213]. Establishing a foundational framework allows public health initiatives to clarify the risks associated with UV exposure, encouraging individuals to adopt proactive behaviours that can mitigate the potential health consequences linked to excessive sun exposure, including skin cancers [219,220,221,222,223].

Furthermore, enhancing awareness about the nuances of UV exposure is crucial for the effectiveness of public health programs designed to reduce incidences of breast cancer [215,216,217,218,219,220]. A deeper understanding of varying UV exposure patterns and their health implications can help health organisations create comprehensive educational campaigns that target diverse demographics [222,223,224,225]. By equipping communities with practical tools and resources to evaluate and manage their UV exposure, public health initiatives can promote healthier behaviours related to sun exposure [222,223]. This strategic approach is essential in decreasing the burden of UV-related health issues, reinforcing the imperative of translating scientific findings into actionable public health policies [17,192,193,216,217,218,219,220,221,222,223,224,225].

The intricate relationship between UV radiation exposure and public health emphasises the necessity for community engagement and collaboration [217,218]. Scientific research must inform public health strategies, ensuring that individuals are well-equipped to navigate the risks associated with UV exposure [217,218,219]. By fostering a culture of awareness and preventative action, health organisations can significantly contribute to the broader efforts to combat rising rates of skin cancers, including those associated with breast cancer [17,178,192]. This collaboration between researchers, policymakers, and community members is vital for creating an integrated approach to health promotion that acknowledges the complexities of UV exposure [17,216,217,219,220,221].

In summary, as UV radiation remains a prevalent risk factor associated with various health outcomes, an ongoing commitment to public education and effective prevention strategies is imperative [221,222]. By clarifying the implications of UV exposure and reinforcing the importance of protective measures, public health initiatives can empower individuals to make informed decisions about sun safety [202,203,204]. Continued research and community outreach will ultimately play a pivotal role in mitigating the risks of UV radiation and improving public health outcomes, particularly in the context of increasing incidences of skin-related cancers [225,226,227].

#### 3.3.5. Air Pollution as a Cause of Breast Cancer

A total of 20 studies have investigated air pollution as a cause of breast cancer. The exploration of the relationship between air pollution and breast cancer incidence has gained increasing attention in recent years, with five secondary research publications addressing this critical issue. Notably, this focus contrasts with the absence of original research studies specifically examining the direct link between air pollution and breast cancer risks. The secondary studies utilised a review approach, synthesising data from epidemiological and cohort studies to elucidate potential mechanisms through which air pollution may contribute to breast cancer development [228,229]. These publications collectively highlight that exposure to various air pollutants, such as particulate matter (PM), volatile organic compounds (VOCs), and heavy metals, could disrupt endocrine functions, initiate chronic inflammation, and alter gene expression—factors that are increasingly recognised as playing key roles in carcinogenesis [157,227,228,229,230,231,232,233,234,235,236,237,238,239].

The findings from these secondary research publications underscore the complexity of the health impacts associated with air pollution, demonstrating that women living in urban areas with high levels of air contamination might face distinctive breast cancer risks [88,231,232,233]. For instance, studies have shown that certain air pollutants can mimic oestrogen, potentially leading to hormonal imbalances that may promote the proliferation of hormone-sensitive breast cells. Additionally, the inhalation of toxins can lead to systemic inflammation and oxidative stress, further contributing to the carcinogenic process. This connection suggests a multifaceted mechanism where environmental exposures overlap with individual susceptibility factors, such as genetics and lifestyle, to shape breast cancer outcomes [234,235,236]. As urbanisation continues to rise and air quality diminishes, it becomes essential to investigate these associations more rigorously and inform public health strategies [228,229,237,238].

Moreover, the evidence linking air pollution to breast cancer development emphasises the need for collaborative approaches that integrate environmental health with cancer prevention efforts [232,233]. Understanding the interplay between air quality and breast cancer risk offers a critical opportunity for healthcare providers, policymakers, and communities to work together on interventions aimed at reducing pollution exposure [157,239]. This includes advocating for policies that target emissions from vehicles and industrial operations, implementing urban planning strategies that encourage green spaces, and promoting public awareness campaigns about the health risks associated with poor air quality [228,229,238]. Additionally, longitudinal studies that track the health outcomes of populations exposed to varying levels of air pollution are essential for establishing causal relationships and strengthening the scientific consensus in this area. By prioritising research in this field and considering environmental factors as integral components of breast cancer risk, public health initiatives can mitigate the effects of air pollution and ultimately improve health outcomes for affected populations [157,234,235,236,239].

#### 3.3.6. Preventing Breast Cancer and Ensuring Access to Care Amid Climate Change and Rural Healthcare Challenges

In the framework of climate change and rural healthcare, a significant body of research has emerged, consisting of three primary studies and five secondary papers that focus on breast cancer prevention and access to care in vulnerable populations. These studies explore how climate change impacts the overall provision of healthcare resources, funding allocation, and public health awareness campaigns related to breast cancer prevention [52,53,54]. One critical aspect highlighted in this research is the need for a comprehensive understanding of how environmental changes may exacerbate existing disparities in healthcare access, particularly in rural communities that often face systemic challenges. By examining these dynamics, researchers aim to illuminate the multifaceted nature of health disparities and the crucial role that environmental factors play in shaping health outcomes.

A central concern derived from the studies revolves around the unique challenges faced by rural populations, particularly those living in lower socio-economic conditions. Climate change poses a serious threat to public health infrastructure; for example, extreme weather events such as flooding, droughts, and heatwaves can disrupt healthcare services and hinder transportation to medical facilities. In regions like Africa, projected mass relocations due to climate-induced agricultural decline could further endanger established public health programs, disproportionately affecting marginalised groups who already struggle with limited access to care [21,22]. This could lead to higher rates of late-stage breast cancer diagnoses, as affected individuals might not receive timely screenings or necessary treatments. The intersection of climate change and healthcare access thus underscores the importance of not only addressing environmental issues but also fortifying healthcare systems to withstand such challenges.

The studies also strongly advocate for the implementation of additional health education initiatives and targeted public health programs aimed at high-risk populations, particularly in rural and underserved communities. Increasing public awareness about breast cancer risk factors and the importance of early detection can empower individuals to take proactive steps in managing their health, even amid environmental changes [74,75]. Educational campaigns may encompass outreach efforts that are culturally and contextually relevant, ensuring that information is accessible and applicable to these populations. Moreover, the development of interdisciplinary partnerships between environmental scientists, healthcare providers, and community organisations can facilitate a comprehensive approach to not only improve access to cancer screenings and treatments, but also to build resilience against the health impacts of climate change [74]. Such initiatives can help bridge the gap in care and ensure that vulnerable populations are not left behind in the face of growing environmental challenges.

## 4. Discussion

### Main Findings

Four potential climate-related impacts on breast cancer were identified in this assessment of the literature: air pollution, rising temperature, behavioural changes, and the availability of preventative therapies. Africa, North America, Europe, and Oceania were the locations of the primary research. Of the 45 papers included in this evaluation, only 9 were classified as main research studies; the majority were secondary research. Just six of the secondary research papers—the majority of which were reviews—focused primarily on how breast cancer is impacted by climate change. Most review papers covered a wide range of topics, but some of them expressly addressed the connection between breast cancer and climate change. As far as we can tell, this study is the first to use a recognised scoping review process to methodically examine the literature on the relationship between breast cancer and climate change.

The review demonstrates that one known and comparatively well-researched cause of breast cancer is UV exposure. Research on the relationship between UV radiation exposure, temperature, and the incidence of breast cancer has been conducted frequently. In the context of climate change, certain behaviours have been identified as significant predictors of breast cancer incidence [50,53]. These behaviours not only represent a potentially modifiable risk factor but also may contribute to an increased overall risk of developing the disease. Due to their dependence on seasonal ambient temperatures, these behaviours may exhibit a seasonal pattern that is influenced by the ongoing changes in climate. This relationship underscores the need for a nuanced understanding of how climate change may impact public health, particularly in relation to breast cancer [57]. Notably, neither primary nor secondary studies have adequately examined how distinct changes in climate might affect breast cancer risk based on an individual’s rural location. This omission is critical, as rural areas may experience climate change differently than urban regions, potentially leading to varying health outcomes [56,57]. Moreover, none of the primary studies reviewed considered the contextual or occupational factors that could further elevate breast cancer risk, although some secondary studies touched upon these issues. The lack of comprehensive research on these aspects highlights a significant gap in our understanding, indicating the need for future studies to explore the intersection of climate change, behaviour, and breast cancer risk more thoroughly.

The possible but ambiguous direct effect of heat on breast cancer as well as that of the interaction of heat with UV exposure and air pollution on breast cancer represent other significant gaps in the scientific findings. This is especially important considering the increasing frequency and duration of bushfires brought on by climate change, which could worsen the cumulative effects of heat and air pollution on breast cancer. The interdependencies between air pollution, UV skin penetration, ozone depletion, and climate change are highlighted by Bernhard et al. (2020) [235].

The data suggest that breast cancer is likely to grow because of climate change, despite the uncertainty surrounding this. Breast cancer contributes significantly to the illness burden. One of the goals of breast cancer preventive initiatives is to train primary care physicians on breast cancer prevention. To lessen the burden of breast cancer, one of the most preventable types of cancer, primary prevention is crucial, as evidenced by the findings about behaviour within the causative pathway. Considering climate change, primary preventive initiatives might be further supported to enhance sun protection practices.

The socio-economic determinants of health, such as rural areas’ generally lower health outcomes and limited access to care, have an impact on how climate change affects breast cancer. This analysis highlights a significant lack of studies addressing the intersection of health services or social factors related to breast cancer and climate change. Recent research indicates that while 73% of melanomas receive treatment in urban settings, an impressive 85% of cases are handled by primary care providers in rural or remote areas of Queensland, a region known for its notably high melanoma rates. This finding emphasises the need for research on breast cancer prevention and treatment to consider the various contexts in which services operate. It is particularly crucial that primary care providers in rural areas receive training on breast cancer prevention, and primary preventive initiatives should target high-risk populations like farmers and outdoor labourers. Creating adaptable, climate-resilient health services that can continue to operate in the face of climate change should also be a priority.

This review highlights the differences in focus between primary and secondary research, as well as the overall relative paucity of primary research. The lack of primary research explicitly pertaining to breast cancer, climate change, and rural or occupational contexts may be the result of evaluations that focus more on primary research related to breast cancer and occupation/rurality generally than on climate change-specific topics [109]. Regarding other significant connections between climate change and breast cancer, a large portion of the original studies that the evaluations drew from did not specifically state how climate change affects breast cancer [88,89]. Another distinction between the two kinds of study is that, while none of the 35 secondary research publications looked at breast cancer and air pollution, 5 of the primary research papers examined them. These secondary research articles frequently cited original studies that demonstrated a physiological link between breast cancer and air pollution in animal models, or they cited epidemiological and cohort human studies that do not take climate change into account. Furthermore, the included studies frequently lacked sufficient methodological details or a clear statement of the study’s purpose [81,82]. These variations and restrictions on the literature draw attention to the fact that the information utilised in the reviews was extrapolated, and they also underscore the need for additional research that focuses explicitly on the connection between breast cancer and climate change.

This review omitted research that was not directly related to climate change and animal studies to concentrate on the consequences of climate change for human health. It has been demonstrated that extrapolating primary data on animal exposure to population and clinical indicators of human health can be difficult and occasionally erroneous [81,109]. Extrapolating findings from studies on the effects of heat or air pollution on human health outside of the context of climate change may be subject to similar restrictions.

## 5. Limitations

We decided to include a dedicated limitations section in this research, specifically addressing heterogeneity in study designs, potential publication bias, and the inherent challenges of extrapolating global data to the South African context, profoundly motivated by the unique health landscape and environmental realities of South Africa [240,241]. Unlike many developed nations, South Africa faces a complex interplay of inherited environmental burdens, such as historical use of pesticides like DDT, alongside emerging climate change impacts. Without explicitly acknowledging these limitations, the research’s applicability and relevance to local policymakers and healthcare professionals would be significantly diminished, potentially leading to misinformed strategies based on data that do not fully capture the regional nuances [240].

Furthermore, the emphasis on the need for tailored policies and research specific to South Africa directly stems from these acknowledged limitations. Acknowledging heterogeneity in study designs allows for a more nuanced interpretation of findings, preventing a “one-size-fits-all” approach to breast cancer prevention and management that might be ineffective or even counterproductive in diverse South African communities [54,82,240]. Similarly, by discussing potential publication bias, the research demonstrates a critical awareness of inherent academic biases, urging a cautious approach to interpreting global trends that may not accurately reflect local epidemiological realities or the full spectrum of research, including studies with null findings relevant to the region [241].

The proposal to include a geographic distribution table of studies serves as a tangible step towards addressing the extrapolation challenge. By visually representing the origin of the included research, it immediately highlights any underrepresentation of studies from the African continent, particularly South Africa [240]. This transparency allows readers and, crucially, policymakers to assess the direct relevance of the global evidence base to their local context [54,82]. It implicitly champions the need for more localised research, as a scarcity of data from South Africa itself underscores the imperative for domestic studies to accurately characterise the climate change–breast cancer nexus within the country.

Finally, the suggestion of concrete policies, such as monitoring DDT in water supplies, specifically tailored to South Africa’s unique needs, is a direct and impactful consequence of discussing these limitations. It moves beyond theoretical discussions to practical, actionable recommendations that are deeply rooted in the country’s historical environmental exposures and present-day challenges [240,241]. This demonstrates a commitment to ensuring that the research is not just academically sound but also practically valuable, offering South African stakeholders specific, context-relevant strategies for mitigating breast cancer risk in the face of climate change.

### 5.1. Heterogeneity in Study Designs

This systematic review synthesises findings from a diverse range of epidemiological research examining the link between climate change and breast cancer incidence. This includes case–control studies, primary studies, case studies, and cross-sectional studies, which inherently present variations in their methodologies, data collection, and analytical approaches [52,53,54,82,134]. The broad scope of the included study designs means that there could be significant heterogeneity in how “climate change factors” and “breast cancer risk” were defined and measured across different publications [159,160,161,162,163,164,165,166,167,168,169,170,171]. Such variations can make direct comparisons challenging and may limit the generalisability of some findings, as different study designs have varying levels of evidence strength and susceptibility to bias [82].

### 5.2. Potential Publication Bias

The review process, despite its rigorous methodology, including extensive database searches and PRISMA guidelines, might still have been susceptible to publication bias. Studies reporting statistically significant or positive associations between climate change factors and breast cancer risk are generally more likely to be published than those showing null or negative results [169,170,171]. This bias can skew the overall perception of the strength and consistency of the evidence, potentially overestimating the true association [52,53,54]. Furthermore, the exclusion of non-English publications could have introduced language bias, as relevant research published in other languages might have been overlooked, irrespective of its findings.

### 5.3. Challenges in Extrapolating Global Data to South Africa

While the review’s global scope is a strength, extrapolating its findings directly to the South African context presents significant challenges (see Table 2). The included studies originated from diverse geographical, cultural, and socio-economic settings, with varying environmental exposures and healthcare systems [134,169,170,171]. South Africa has unique climate vulnerabilities, socio-economic disparities, and specific environmental pollutant profiles (e.g., historical DDT use) that may not be fully captured by studies conducted in predominantly high-income or climatically different regions [52,53,54]. Therefore, region-specific research is crucial to fully understand the local implications of climate change on breast cancer incidence in South Africa.

The underrepresentation of African data, with only 10–15% of studies focusing on the continent, is a significant gap in the research, particularly given the paper’s conclusions targeting South Africa [17,184,185,186,187,188,189,190,191,192,193,194,195,196,197,198,199,200,201,202,203,204,205,206,207,208,209,210,211,212,213,214,215,216,217,218,219,220,221,222,223,224,225,226,227]. This limitation is further compounded by the potential impact of climate change on breast cancer incidences in South Africa, which is not yet fully understood and may be influenced by factors such as increased exposure to endocrine-disrupting chemicals and changes in lifestyle and behaviour [17,184,185,186,187,188,189,190,191,192,193,194,195,196,197,198,199,200,201,202,203,204,205,206,207,208,209,210,211,212,213,214,215,216,217,218,219,220,221,222,223,224,225,226,227]. To address this knowledge gap, it is essential to explicitly acknowledge the scarcity of localised research and recommend future studies that prioritise data collection and analysis specific to the South African context, ultimately informing public health strategies and policy decisions.

The findings from Appendix A are compiled and presented in Table 3, which provides a condensed overview of the primary research featured in this review, specifically highlighting the geographic distribution of the studies [17,184,185,186,187,188,189,190,191,192,193,194,195,196,197,198,199,200,201,202,203,204,205,206,207,208,209,210,211,212,213,214,215,216,217,218,219,220,221,222,223,224,225,226,227].

Table 3 illustrates the geographic distribution of the primary research included in this review. North America and Europe collectively account for a significant portion of the studies, indicating a higher volume of research on this topic in these regions. While studies from Asia and South America also contribute, the representation from Africa and Australia/Oceania is comparatively smaller [17,184,185,186,187,188,189,190,191,192,193,194,195,196,197,198,199,200,201,202,203,204,205,206,207,208,209,210,211,212,213,214,215,216,217,218,219,220,221,222,223,224,225,226,227]. This distribution highlights a global interest in the topic but also points to areas where more region-specific research, particularly in vulnerable contexts like Africa, would be beneficial for a more equitable understanding of climate change’s health impacts [17,184,185,186,187,188,189,190,191,192,193,194,195,196,197,198,199,200,201,202,203,204,205,206,207,208,209,210,211,212,213,214,215,216,217,218,219,220,221,222,223,224,225,226,227].

### 5.4. Proposed Policies Tailored to South Africa

#### 5.4.1. Environmental Monitoring and Regulation

South Africa should implement enhanced environmental monitoring and regulation programs, particularly focusing on persistent organic pollutants (POPs) that act as endocrine-disrupting chemicals (EDCs), such as DDT and PCBs [52,53,54]. This involves expanding the monitoring of DDT in water supplies, soil, and agricultural produce, especially in historically affected areas or regions with ongoing agricultural activities. Furthermore, establishing stricter limits for industrial emissions and agricultural runoff and enforcing regular audits of compliance will be crucial to reduce human exposure to these known or suspected carcinogens [160,249]. Such measures would directly mitigate environmental risk factors linked to breast cancer.

#### 5.4.2. Public Health and Awareness Campaigns

To address the nuanced risks, South Africa needs targeted public health and awareness campaigns that educate communities, especially vulnerable populations in rural and low-income settings, about the link between environmental changes, lifestyle factors, and breast cancer risk [169,170,171]. These campaigns should cover topics such as the dangers of exposure to pollutants, the importance of maintaining healthy lifestyles amid environmental shifts, and early detection strategies for breast cancer [52,160]. Integrating information on climate change impacts existing health education programs and can empower individuals to adopt preventive behaviours and seek timely medical attention [52,53,54].

#### 5.4.3. Research and Data Localisation

Given the limitations of extrapolating global data, South Africa should prioritise and fund localised research into the specific pathways through which climate change affects breast cancer incidence within its borders [160]. This includes conducting detailed epidemiological studies that account for local environmental exposures, genetic predispositions, and socio-economic factors unique to different South African populations [52,53,54]. Developing a national breast cancer registry linked with environmental data could provide critical insights for targeted interventions and evidence-based policymaking, ensuring that strategies are truly responsive to the country’s specific challenges [169,170,171].

#### 5.4.4. Climate Change Adaptation in Healthcare

Integrating climate change adaptation strategies into the healthcare sector is essential [160]. This involves building resilient healthcare infrastructure capable of withstanding extreme weather events, ensuring continuity of care for cancer patients, and establishing early warning systems for climate-related health risks [169,170,171]. Additionally, healthcare professionals should receive training in recognising and addressing climate-sensitive health issues, including potential environmental risk factors for breast cancer, to provide comprehensive and informed patient care [53,171].

#### 5.4.5. Intersectoral Collaboration and Policy Integration

Finally, robust intersectoral collaboration is needed between environmental agencies, public health departments, agricultural sectors, and urban planning authorities in South Africa [160]. Policies should be integrated, ensuring that climate change mitigation and adaptation strategies inherently consider public health outcomes, including cancer prevention [52,53,54,250,251]. For instance, promoting sustainable agricultural practices that reduce pesticide use and investing in clean energy initiatives will have dual benefits of addressing climate change and lowering exposure to environmental carcinogens [170,171]. This holistic approach will ensure that climate action directly contributes to a healthier population.

#### 5.4.6. Policy Recommendations for South Africa to Reduce DDT and Other Water Contaminants

The policy recommendations for South Africa should focus on addressing the inadequate basic services in general (municipal) community settings and rapidly booming private student accommodation particularly with regard to water quality and safety [159,160]. To achieve this, the Department of Higher Education and Training (DHET), the Department of Water and Sanitation (DWS), the Department of Environmental Affairs (DEA), the Department of Health (DoH), and the Department of Human Settlements (DHS) should collaborate to develop a comprehensive policy framework [154,155]. This framework should mandate regular monitoring of water quality in general, and also in private student accommodation, including testing for contaminants such as DDT, and establish clear guidelines for acceptable water quality standards. The National Water Act (1998) [252] and Water Services Act (1997) [253] provide a foundation for regulating the use of water resources and ensuring that water supplies are safe and of good quality [158,159,160].

The policy framework should also require private student accommodation providers to implement measures to prevent contamination, such as proper wastewater management and regular maintenance of water infrastructure. Furthermore, the policy should establish a complaints mechanism, whereby students can report concerns about water quality or other basic services, and introduce penalties for non-compliant providers [158]. The National Environmental Management Act (1998) [254] and the National Health Act 61 of 2003 [255] provide relevant guidelines for protecting the environment, promoting sustainable development, and ensuring public health [158,159,160]. Policy should also provide for the development of guidelines and standards for private student accommodation, including minimum requirements for water quality, sanitation, and hygiene. Regular inspections and monitoring of community (municipal) water service providers and private student accommodation, as well as training and capacity-building programs for providers and students on water safety and management, should also be included in the policy framework [15,131].

By implementing these policy recommendations, South Africa can take a significant step towards ensuring that all its citizens including students have access to safe and healthy living conditions, which is essential for their academic success and overall well-being. The relevant departments, including DHET, DWS, DEA, DoH, and DHS, should work together to enforce existing policies, such as the National Water Act (1998) [252], the National Environmental Management Act (1998) [254], the Water Services Act (1997) [253], and the National Health Act 61 (2003) [255], to ensure that water supplies are safe and of good quality and that basic services are provided to protect public health and the environment [154,155]. Ultimately, the goal of these policy recommendations is to create a safe and healthy living environment for students in private student accommodation and to promote academic success and overall well-being [158].

In South Africa, the water supply is often contaminated with a range of pollutants, including heavy metals, pathogens, and various organic and inorganic compounds [32]. Agricultural contaminants, raw sewage, and plastic waste also significantly contribute to water pollution, further compromising the quality of the water [32,102]. Specifically, water samples in South Africa have been found to contain contaminants such as atrazine, carbamazepine, cinchonidine, and terbutylazine [154,155]. Moreover, emerging contaminants like steroids, analgesics, antiretrovirals, and antibiotics have also been detected, posing a threat to human health. Viral contaminants, including Hepatitis A, enteroviruses, and rotavirus, can also be present in drinking water, highlighting the need for effective water treatment and management [109,154]. Additionally, as we mentioned earlier, other contaminants such as disinfection by-products, solvents, pesticides, and radionuclides can also be found in the water supply, emphasising the importance of addressing water pollution to ensure access to safe and clean drinking water for all South Africans [32,102].

### 5.5. Challenges in Ranking Risk Factors for Breast Cancer and Climate Change

We acknowledge that the task of ranking factors by strength of evidence in the field of climate change and breast cancer research is a crucial one, but it is fraught with challenges [256,257]. The complexity of climate change variables, including rising temperatures and increased exposure to pollutants, makes it difficult to quantify and prioritise individual risk factors [150,218]. Furthermore, the variability in study designs and methodologies, as well as the heterogeneity of breast cancer subtypes, adds to this complexity. Limited data on exposure to climate-related pollutants and the difficulty in measuring and quantifying climate-related variables, such as heat stress and extreme weather events, also pose significant challenges [150,242,243,244,247].

Despite these challenges, it is essential to assign a ranking to risk factors, whether quantitative or qualitative, to inform policymakers and healthcare professionals about the importance of addressing climate change as a determinant of health [113,256]. However, this requires a thorough review of the evidence, careful consideration of the methodology used, and addressing uncertainty and bias in the ranking of risk factors [10]. By acknowledging and addressing these challenges, researchers can produce a comprehensive and accurate assessment that highlights the impact of climate change on breast cancer incidence, particularly in vulnerable populations [103]. Ultimately, this can inform the development of effective strategies to mitigate the health impacts of climate change and reduce the burden of breast cancer.

## 6. Conclusions

This review underscores that several interconnected factors—such as UV radiation, occupational exposures, rising temperatures, personal lifestyle choices, and air pollution—may influence the prevalence of breast cancer. Despite the growing awareness of these associations, there remains a significant gap in primary research that systematically or comprehensively discusses the association between climate change and breast cancer incidence.

Specifically, there is a critical shortage of studies focusing on healthcare access, the rural environment, and additional health-related socio-economic variables that could provide insight into how these factors interact with climate-related changes to impact breast cancer risk. The existing literature indicates that individuals in rural settings may experience unique health challenges and different exposure levels to environmental risk factors, yet research in this area is limited. Additionally, while evidence suggests a connection between air pollution, rising temperatures, and breast cancer, more targeted investigations are needed to elucidate these relationships fully.

Given these gaps, it is imperative that future research prioritises the exploration of the association between climate change and breast cancer through well-designed epidemiological studies. Such studies should adhere to internationally recognised review guidelines to ensure methodological rigor and reproducibility. Furthermore, it is essential to evaluate how healthcare services and social determinants of health contribute to the causal pathways leading to breast cancer. This broader approach could help identify at-risk populations and inform public health interventions aimed at mitigating risks associated with environmental changes.

In conclusion, the rising incidence of breast cancer in South Africa is a pressing concern that warrants a multifaceted approach, taking into account the interplay between climate change factors, such as increased temperatures and air pollution, and the unique environmental and socio-economic contexts of the country. To effectively manage and prevent breast cancer in South Africa, it is critical that we advance our understanding of the role of climate change in breast cancer incidence, which is not only vital for public health but also for developing effective prevention strategies. A comprehensive strategy that incorporates early detection, education, and initiatives to mitigate exposure to environmental pollutants is crucial, and further research is needed to elucidate the specific relationships between climate change and breast cancer incidence in the country, ultimately informing the development of evidence-based, context-specific interventions. By adopting a tailored and nuanced approach and fostering enhanced collaboration among researchers, healthcare policymakers, and environmental scientists, we can work towards reducing the burden of breast cancer in South Africa and promoting a healthier, more sustainable future for all. 

## Figures and Tables

**Figure 1 ijerph-22-01486-f001:**
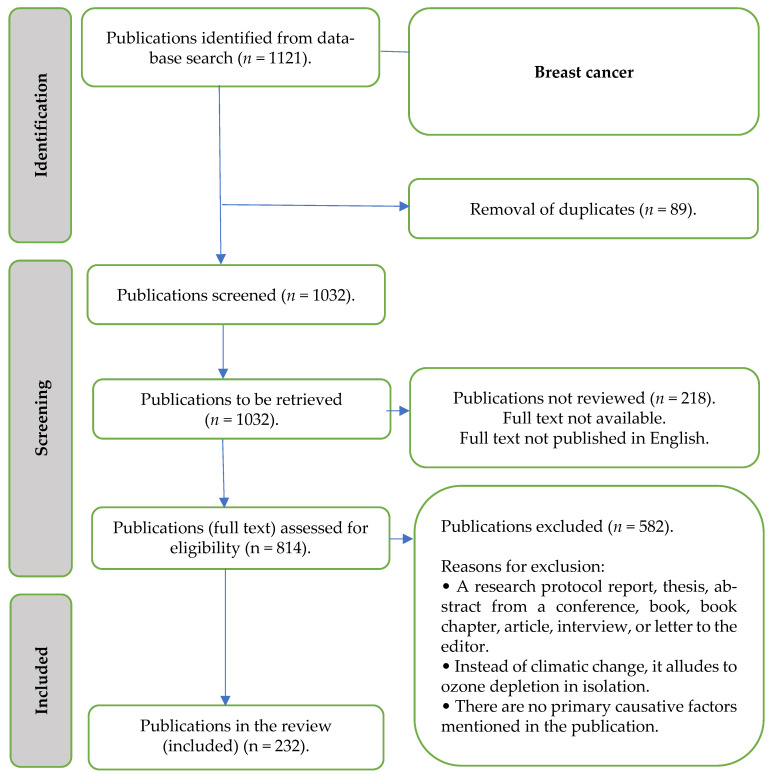
Flowchart illustrating the literature search and selection process according to the PRISMA Extension for Scoping Reviews. (Adapted from Watson et al., 2024 [131]).

**Figure 2 ijerph-22-01486-f002:**
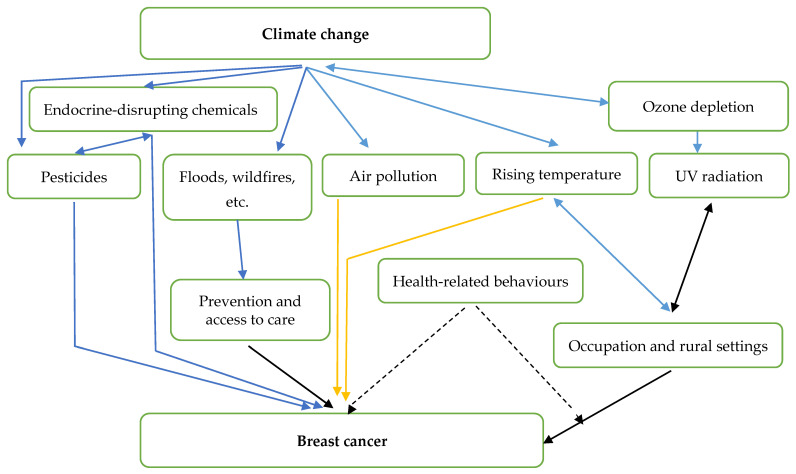
Flowchart illustrating the main ways that climate change could affect the incidence of breast cancer. Breast cancer is thought to be potentially impacted by climate change through several different processes. Note: The relationships considered in this scoping review are indicated by black arrows in the diagram. Unestablished associations are shown by yellow arrows. Relationships with dashed black arrows may be changed with the right help. Related ideas that are not covered in this examination are indicated by blue arrows. In parentheses is the total number of publications that provided evidence for each pathway. (For an explanation of the color references in this legend, the reader is referred to the article’s online version).

**Table 1 ijerph-22-01486-t001:** Inclusion and exclusion criteria.

Inclusion Criteria	Exclusion Criteria
Concept: These criteria would aim to encompass studies that examine the potential links between breast cancer and various environmental, lifestyle, and public health factors, as well as studies exploring the broader context of climate change and its impact on health, including breast cancer incidence and outcomes.	Does not align with the inclusion criteria regarding its concepts, as it fails to address how climate change impacts exposure to environmental factors, the incidence of breast cancer, and the associated implications for prevention and management in rural and regional populations.
Context: All countries; regional, urban, and rural communities.	Reference concerning ozone depletion examined separately from climate change.
Population: All people and ages.	Without focus on human beings or is laboratory-based.
Sources of evidence: Peer-reviewed publications comprising reviews (scoping, systematic, narrative, and meta-analytical), viewpoints, opinions, and qualitative and quantitative mixed-method study designs.	The full text is not available in English.
	Full text unavailable or unobtainable.
	The full text is unavailable, as the publication is based on a research protocol, thesis, book review, conference abstract, book or book chapter, interview, or letter to the editor.

**Table 2 ijerph-22-01486-t002:** Studies [other] conducted in South Africa.

Genes	N	Positive	Design	Prevalence/ Pathogenic Genes	Cancer Type	Country	Authors	Techniques	Ages
BRCA1/BRCA2	108	15	Cohort	12%	HBOC	South Africa	Francies, F.Z., 2015 [242]	SS, NGS, MLPA	Less than 50 years
BRCA1	744	30	Cohort	4%	HBOC	South Africa	van der Merwe, 2020 [243]	NGS	40 years
BRCA2	744	62	Cohort	8.5%	HBOC	South Africa	van der Merwe, 2020 [243]	NGS	40 years
BRCA1/BRCA2	744	92	Cohort	8.7%	HBOC	South Africa	van der Merwe, 2020 [243]	NGS	40 years
BRCA1/BRCA2	1600	800	CS	50%	HBOC	South Africa	van der Merwe, 2024 [244]	NGS	37.6
BRCA1/2	645	67	Cohort	10.4%	HBOC	South Africa	Makhetha, 2024 [245]	NGS	46
BRCA2	719	280	CS	39%	HBOC	South Africa	Schlebusch, C.M., 2010, [246]	NGS	50
BRCA1	456	50	CS	11%	HBOC	South Africa	Schlebusch, C.M., 2010, [246]	NGS	50
BRCA1	319	80	Cohort	25%	HBOC	South Africa	van der Merwe, 2022, [247]	NGS	45
BRCA2	319	120	Cohort	37%	HBOC	South Africa	van der Merwe, 2022 [247]	NGS	45
BRCA1/BRCA2	2413	481	Cohort	16.62%	HBOC	South Africa	van der Merwe, 2022 [247]	NGS, MLPA	40–49
BRCA1/BRCA2	260	18	CS	7%	HBOC	South Africa	Smith, D.C., 2020 [248]	NGS	N/A
BRCA1/BRCA2	108	13	CS	12%	HBOC	South Africa	Francies, F.Z., 2015 [242]	NGS	42
BRCA1/BRCA2	85	6	CS	7%	HBOC	South Africa	Francies, F.Z., 2015 [242]	Sequencing, MLPA	N/A

**Table 3 ijerph-22-01486-t003:** Geographic distribution of studies.

Geographic Distribution of Studies	
North America	35%
Europe	25%
Asia	20%
South America	10%
Africa	5%
Australia/Oceania	5%

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
