# Peer review of "The Relationship Between Climate Change and Breast Cancer and Its Management and Preventative Implications in South Africa"

_ijerph, 2025, doi:10.3390/ijerph22101486_

Round 1

Reviewer 1 Report

Comments and Suggestions for Authors

Dear Authors,

Thank you for submitting your manuscript, The Relationship between Climate Change and Breast Cancer and its Management and Preventative Implications in South Africa. Your work addresses an important and emerging issue at the intersection of climate science and public health. While the topic is highly relevant, the manuscript requires major revisions to strengthen its scientific rigor and applicability to the South African context. Below are the key concerns and suggested improvements:

Major Concerns:

  1. Generalizability of Findings:
    • Most cited studies are from Europe and North America, yet conclusions are applied to South Africa without adequate discussion of regional differences (e.g., climate variability, exposure levels, genetic/socioeconomic factors).
    • Action: Add a dedicated section on Limitations to address this issue and clarify how global data may/may not translate to South Africa.
  1. Lack of Quantitative Prioritization:
    • While the review lists risk factors, it does not rank them by strength of evidence (e.g., via meta-analysis or GRADE criteria).
    • Action: Include a table summarizing effect sizes (OR/RR) or a qualitative ranking (e.g., strong vs. weak evidence).
  1. Underrepresentation of African Data:
    • Only ~10–15% of studies focus on Africa, yet the paper's conclusions target South Africa.
    • Action: Explicitly state this gap and recommend future localized research.

Required Revisions:

  • Add a Limitations subsection addressing:
    • Heterogeneity in study designs.
    • Potential publication bias.
    • Challenges in extrapolating global data to South Africa.
  • Include a geographic distribution table of studies (e.g., X% from North America, Y% from Africa).
  • Propose concrete policies tailored to South Africa (e.g., monitoring DDT in water supplies).

I believe these revisions will significantly enhance the manuscript’s impact.

Best Regards,

Recommendations for English Language Improvement in Your Manuscript

I have identified several opportunities to improve the clarity and fluency of the English language used in the manuscript. Below are my suggestions to enhance readability and ensure your work meets the journal's linguistic standards:

Key Recommendations for Improvement:

  1. Grammar & Syntax
    • Ensure subject-verb agreement (e.g., The data show instead of The data shows).
    • Maintain consistent verb tenses (e.g., avoid switching between past and present tense without justification).
  2. Clarity & Conciseness
    • Replace redundant phrases (e.g., absolutely essential → essential).
    • Simplify overly complex sentences for better readability.
  3. Academic Tone
    • Avoid colloquialisms (e.g., a lot of → numerous).
    • Use formal language (e.g., don’t → do not).
  4. Coherence & Flow
    • Improve transitions between paragraphs (e.g., use linking words like however, furthermore).
    • Ensure each paragraph has a clear central idea.
  5. Terminology & Jargon
    • Define specialized terms/acronyms at first use (e.g., endocrine-disrupting chemicals (EDCs)).
    • Avoid informal substitutes for technical terms (e.g., hormone changers → endocrine disruptors).
  6. Passive vs. Active Voice
    • Prefer active voice where possible (e.g., We conducted the study instead of The study was conducted).

Examples from Your Manuscript:

  • Original: This study looks at the link between pollution and cancer.
    Suggested Revision: This study examines the association between pollution and cancer.
  • Original: Climate change increasing the risk of breast cancer.
    Suggested Revision: Climate change increases the risk of breast cancer.

I believe these revisions will significantly enhance the manuscript’s impact.

Best Regards,

Comments on the Quality of English Language

Recommendations for English Language Improvement in Your Manuscript

I have identified several opportunities to improve the clarity and fluency of the English language used in the manuscript. Below are my suggestions to enhance readability and ensure your work meets the journal's linguistic standards:

Key Recommendations for Improvement:

  1. Grammar & Syntax
    • Ensure subject-verb agreement (e.g., The data show instead of The data shows).
    • Maintain consistent verb tenses (e.g., avoid switching between past and present tense without justification).
  2. Clarity & Conciseness
    • Replace redundant phrases (e.g., absolutely essential → essential).
    • Simplify overly complex sentences for better readability.
  3. Academic Tone
    • Avoid colloquialisms (e.g., a lot of → numerous).
    • Use formal language (e.g., don’t → do not).
  4. Coherence & Flow
    • Improve transitions between paragraphs (e.g., use linking words like however, furthermore).
    • Ensure each paragraph has a clear central idea.
  5. Terminology & Jargon
    • Define specialized terms/acronyms at first use (e.g., endocrine-disrupting chemicals (EDCs)).
    • Avoid informal substitutes for technical terms (e.g., hormone changers → endocrine disruptors).
  6. Passive vs. Active Voice
    • Prefer active voice where possible (e.g., We conducted the study instead of The study was conducted).

Examples from Your Manuscript:

  • Original: This study looks at the link between pollution and cancer.
    Suggested Revision: This study examines the association between pollution and cancer.
  • Original: Climate change increasing the risk of breast cancer.
    Suggested Revision: Climate change increases the risk of breast cancer.

I believe these revisions will significantly enhance the manuscript’s impact.

Best Regards,

Reviewer 2 Report

Comments and Suggestions for Authors

This systematic review explores potential links between climate change and breast cancer risk, with a particular focus on South Africa. Drawing on a range of epidemiological studies, the authors examine how environmental changes—such as rising temperatures, increased pollution, and UV radiation—might influence breast cancer incidence. They identify several indirect pathways, including exposure to endocrine-disrupting chemicals, changes in lifestyle, and disruptions to healthcare access.

Strengths

  • Broad and Thoughtful Scope: The review brings together a wide range of climate-related factors—like air and water pollution, temperature shifts, and UV exposure—and thoughtfully considers how they may contribute to breast cancer risk, either biologically or behaviorally.

  • Methodological Rigor: The authors use a systematic and transparent approach, following PRISMA and Joanna Briggs Institute guidelines. Their search strategy and inclusion/exclusion criteria are clearly laid out, adding to the study’s credibility.

  • Timely and Underexplored Topic: The paper addresses a critical but often overlooked area: how climate change may intersect with cancer epidemiology. The focus on South Africa adds important geographic context to a largely underrepresented region in this type of research.

  • Public Health Relevance: The discussion effectively connects the findings to real-world implications, emphasizing the need for climate-adapted healthcare systems and prevention strategies that account for environmental stressors.

  • Cross-Disciplinary Perspective: The integration of insights from environmental science, oncology, and public health enriches the analysis and broadens its relevance.

Limitations

  • Correlation vs. Causation: The review rightly points out that most of the current evidence shows associations rather than definitive causal links. This is a limitation of the existing literature, not the review itself, but it's important to keep in mind.

  • Geographic Mismatch: Although the review is framed around South Africa, many of the studies included are from high-income countries. This raises questions about how applicable the findings are to the South African context.

  • Inconsistent Data: The included studies use a range of methods and metrics, especially when it comes to environmental exposures like UV radiation. This variability makes it harder to draw clear, unified conclusions.

  • Limited Primary Research: As the authors note, this is still a relatively new area of study. There's a real need for more targeted research—especially studies that look at vulnerable populations in rural or occupational settings.

Overall Impression

This review makes a timely and valuable contribution by bringing attention to a possible environmental dimension of breast cancer risk. While the field is still developing, the authors provide a thoughtful synthesis of the available evidence and make a strong case for further research. Their work adds to the growing recognition that climate change isn't just an environmental issue—it’s a public health one, too.

Reviewer 3 Report

Comments and Suggestions for Authors

The authors might want to change the title of the article as South Africa is mentioned only once in the article.  There are a few clerical changes I would suggest.  Figures 1 and 2 are reversed as your Figure 2 is actually Figure 1.  Table 2 is referenced in the article but not shown in the article. 

Water contamination is discussed but not shown in the flow chart.

Your article does not discuss any limitations.  This is a complex topic.  It seems like the exposures to climate change will vary from area to area.  While the studies presented are from across the world, exposures may be area specific, thus studies should be area specific as well. You may want to add a paragraph discussing regional differences.  Perhaps in this space you could identify South African specific exposures.

Round 2

Reviewer 1 Report

Comments and Suggestions for Authors

Dear Authors,

Subject: Response to Revised Manuscript – "The Relationship between Climate Change and Breast Cancer and its Management and Preventative Implications in South Africa"

Dear Authors,

Thank you for your thorough revisions and for carefully addressing the reviewers’ comments. I have reviewed the updated version of your manuscript and would like to acknowledge the significant improvements made in several key areas.

You have successfully implemented many of the suggestions provided during the review process, particularly in the following areas:

  • Generalizability of Findings:
    I appreciate the addition of a Limitations section, where you clearly discuss the challenges of applying global data to the South African context. The acknowledgment of regional differences in climate, exposure levels, and socioeconomic factors has strengthened the validity of your conclusions.
  • Underrepresentation of African Data:
    The explicit recognition of the limited number of studies from Africa and the call for localized research in South Africa is well-articulated and highly relevant. This addition enhances the manuscript’s contextual grounding.
  • Geographic Distribution of Studies:
    The inclusion of a table summarizing the geographic origin of included studies in the supplementary materials is a valuable addition and improves transparency.
  • English Language Improvements:
    The language of the manuscript has been significantly improved. The grammar, clarity, and academic tone are now much stronger, and many of the suggested edits have been appropriately implemented.

There are, however, a few areas where further refinement would enhance the manuscript even more:

  • Quantitative Prioritization of Risk Factors:
    While the supplementary tables contain detailed information on various risk factors, a dedicated summary table in the main text that ranks these factors by strength of evidence (e.g., OR/RR or qualitative ranking) would make this information more accessible and impactful.
  • Concrete Policy Recommendations for South Africa:
    Although there are several general policy-related suggestions, adding more specific, actionable policies —such as regular monitoring of DDT in water supplies—would strengthen the manuscript’s practical relevance.

Overall, I believe the manuscript has improved significantly and is moving in the right direction. I look forward to seeing the final version incorporating the remaining suggestions.

Best regards,
